# RecFlow: An Industrial Full Flow Recommendation Dataset

**Qi Liu**[1],    **Kai Zheng**[2],    **Rui Huang**[2],    **Wuchao Li**[1],    **Kuo Cai**[2],    **Yuan Chai**[2],

**Yanan Niu**[2],    **Yiqun Hui**[2],    **Bing Han**[2],    **Na Mou**[2],    **Hongning Wang**[4],

**Wentian Bao**[3],    **Yunen Yu**[3],    **Guorui Zhou**[2],    **Han Li**[2],    **Yang Song**[2],    **Defu Lian**[1*]

**Kun Gai**[3]

[1]University of Science and Technology of China    [2]Kuaishou
[3]Independent    [4]Tsinghua University
{qiliu67,liwuchao}@mail.ustc.edu.cn,  {liandefu}@ustc.edu.cn
{zhengkai,huangrui06,caikuo,niuyanan,chaiyuan}@Kuaishou.com
{huiyiqun,hanbing,zhouguorui,lihan08,songyang}@Kuaishou.com
{hw5x}@virginia.edu,  {wb2328}@columbia.edu
{yuenyun}@126.com,  {285208254,gai.kun}@qq.com

## Abstract

Industrial recommendation systems (RS) rely on the multi-stage pipeline to balance effectiveness and efficiency when delivering items from a vast corpus to users. Existing RS benchmark datasets primarily focus on the exposure space, where novel RS algorithms are trained and evaluated. However, when these algorithms transition to real-world industrial RS, they face two critical challenges: (1) handling unexposed items—a significantly larger space than the exposed one, profoundly impacting their practical performance; and (2) overlooking the intricate interplay between multiple stages of the recommendation pipeline, resulting in suboptimal system performance. To bridge the gap between offline RS benchmarks and real-world online environments, we introduce RecFlow—an industrial full-flow recommendation dataset. Unlike existing datasets, RecFlow includes samples not only from the exposure space but also from unexposed items filtered at each stage of the RS funnel. RecFlow comprises 38 million interactions from 42,000 users across nearly 9 million items with additional 1.9 billion stage samples collected from 9.3 million online requests over 37 days and spanning 6 stages. Leveraging RecFlow, we conduct extensive experiments to demonstrate its potential in designing novel algorithms that enhance effectiveness by incorporating stage-specific samples. Some of these algorithms have already been deployed online at KuaiShou, consistently yielding significant gains. We propose RecFlow as the first comprehensive whole-pipeline benchmark dataset for the RS community, enabling research on algorithm design across the entire recommendation pipeline, including selection bias study, debiased algorithms, multi-stage consistency and optimality, multi-task recommendation, and user behavior modeling.

## 1 Introduction

Recommendation systems (RS) are pivotal in modern web and mobile applications that handle vast amounts of information. Their primary objective is to deliver personalized recommendations from an extensive corpus of items, based on estimated user preferences. To meet stringent online latency requirements, industrial RS predominantly employs a multi-stage funnel-like pipeline (Covington et al., 2016), striking a balance between effectiveness and efficiency. Substantial efforts have been

---
*Corresponding author

devoted to designing algorithms within this system, aiming to enhance its effectiveness as measured by user feedback on selected items. A typical multi-stage RS consists of successive stages: **retrieval → pre-ranking → ranking → re-ranking**. During online serving, the retrieval stage (Hidasi et al., 2015; Kang & McAuley, 2018; Zhu et al., 2018) retrieves thousands of preferred items from the entire corpus. The pre-ranking stage (Huang et al., 2013; Wang et al., 2020) filters out less favorable items from the retrieved set, forwarding hundreds of more promising items to the ranking stage. In turn, the ranking stage (Cheng et al., 2016; Zhou et al., 2018; Bian et al., 2022) selects the most appealing items from this refined set. Finally, the re-ranking (Pei et al., 2019; Bello et al., 2018) stage determines the final items to be displayed, considering both diversity and business objectives. Notably, as we progress through the stages, the model complexity tends to increase, incorporating additional features and interleaving them at shallow layers of deep neural network models. Importantly, the latter three stages typically learn from the exposure space, which captures actual user feedback (both positive and negative) on the displayed items.

Despite the maturity of industrial RS, two significant shortcomings persist. First, a discrepancy exists between the data distribution in the training space and that in the serving space (Qin et al., 2022). The former corresponds to the exposed space, while the latter primarily resides in the unexposed space. This discrepancy, which we refer to as the distribution shift problem, poses challenges. For instance, consider the pre-ranking model (Wang et al., 2020): It must score thousands of items, yet only a few of these items are exposed to users and stored as training data in each request. Most of the remaining samples have not been exposed even once. Consequently, a pre-ranking model trained solely on the exposure space may inaccurately predict preferences in the retrieved space, leading to suboptimal recommendations (Wei et al., 2024). Similar issues arise in the ranking and re-ranking stages. Second, there is a discrepancy between the learning and serving environments. Although models at different stages are learned and evaluated separately, they must collaborate as a cohesive system to meet user preferences. Insufficient knowledge about other stages during the learning process can result in suboptimal performance when these learned models serve online. For example, the online performance of a retrieval algorithm not only depends on itself but is also influenced by subsequent stages. Incorporating knowledge from subsequent stages can enhance the retrieval algorithm's performance (Ding et al., 2019; Lou et al., 2022; Zheng et al., 2024).

Large-scale datasets serve as the bedrock for advancing various machine learning algorithms. For instance, ImageNet (Deng et al., 2009) has significantly contributed to computer vision, while GLUE (Wang et al., 2018) has played a crucial role in natural language processing. However, in the RS domain, existing datasets (Harper & Konstan, 2015; Ni et al., 2019; Asghar, 2016; Zhu et al., 2018; Yuan et al., 2022; Gao et al., 2022a;b; Sun et al., 2023)—though instrumental in fueling RS research—have a limitation: they are exclusively collected from the exposure space. These datasets cannot fully capture the true dynamics of online recommendation services. Moreover, this inherent bias prevents them from effectively addressing the discrepancy between training and serving in RS.

We propose RecFlow, an industrial large-scale full-flow dataset collected from the real industrial RS. The industrial RS's multi-stage funnel-like pipeline encompasses the following stages: retrieval, pre-ranking, coarse ranking, ranking, re-ranking, and edge ranking. Unlike all previous RS benchmarks, RecFlow samples representative unexposed items from each stage of the funnel in a single request alongside all the exposed items. The inclusion of full-stage samples in each request provides several merits. (1) By recording items from the serving space, RecFlow enables the study of how to alleviate the discrepancy between training and serving for specific stages during both the learning and evaluation processes (Qin et al., 2022). (2) RecFlow also records the stage information for different stage samples, facilitating research on joint modeling of multiple stages, such as stage consistency or optimal multi-stage RS (Zheng et al., 2024). (3) The positive and negative samples from the exposure space are suitable for classical click-through rate prediction or sequential recommendation tasks (Zhou et al., 2018; Kang & McAuley, 2018). (4) RecFlow stores multiple types of positive feedback (e.g., effective view, long view, like, follow, share, comment), supporting research on multi-task recommendation (Ma et al., 2018a; Zhao et al., 2019; Tang et al., 2020; Liu et al., 2023). (5) Information about video duration and playing time for each exposure video allows the study of learning through implicit feedback, such as predicting playing time (Covington et al., 2016; Lin et al., 2023). (6) RecFlow includes a request identifier feature, which can contribute to studying the re-ranking problem (Pei et al., 2019; Bello et al., 2018). (7) Timestamps for each sample enable the aggregation of user feedback in chronological order, facilitating the study of user behavior sequence modeling algorithms (Zhou et al., 2018; 2019; Chang et al., 2023; Hou et al., 2023). (8) RecFlow

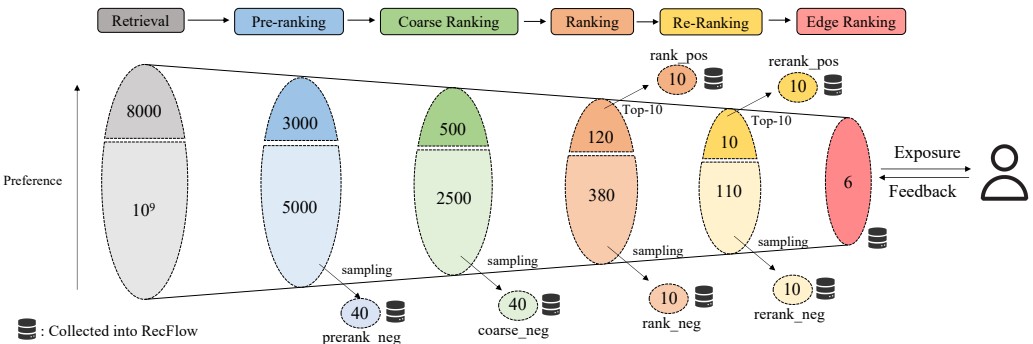

Figure 1: The overall collection process of RecFLow.

incorporates context, user, and video features beyond identity features (e.g., user ID and video ID), making it suitable for context-based recommendation (Huang et al., 2019; Wang et al., 2022). (9) The rich information recorded about RS and user feedback allows the construction of more accurate RS simulators or user models in feed scenarios (Shi et al., 2019; Zhao et al., 2023). (10) Rich stage data may help estimate selection bias more accurately and design better unbiased algorithms (Chen et al., 2023). Furthermore, RecFlow is a large-scale dataset, containing 38 million exposure samples and 1.9 billion stage samples, ensuring the credibility of algorithm improvements based on its data.

Given these characteristics, RecFlow can be utilized across a broad spectrum of RS algorithms. In this paper, we primarily conduct pioneering experiments to explore its potential in each stage of the RS funnel. In the retrieval stage, we investigate the effectiveness of using filtered videos from each stage as hard negative samples and explore the interplay between retrieval and subsequent stages. For the coarse ranking stage, we leverage corresponding stage samples to address the distribution shift problem and model mutual effects between stages. Motivated by existing works, we explore how to exploit stage samples for designing auxiliary ranking tasks and behavior sequence modeling algorithms to improve classical AUC metrics. Similar exploration experiments are also conducted for the ranking stage. Notably, RecFlow also introduces a new recall metric to assess the performance of different methods based on stage samples to mitigate the gap between training and serving environments. RecFlow is the first RS dataset containing stage samples. It stands as one of the largest and most comprehensive datasets for RS, covering nearly all recommendation tasks. We have made the dataset and source codes publicly available to promote reproducibility and advance RS research at https://github.com/RecFlow-ICLR/RecFlow. The dataset is licensed under CC-BY-NC-SA-4.0 International License.

## 2 DATASET CHARACTERISTIC

### 2.1 COLLECTION

RecFlow is the first RS dataset containing intermediate filtered videos of each stage in the industrial RS funnel. The multi-stage funnel-like pipeline of the industrial RS contains six stages, including **retrieval → pre-ranking → coarse ranking → ranking → re-ranking → edge ranking**. The number of videos output at each stage is $8000 \rightarrow 3000 \rightarrow 500 \rightarrow 120 \rightarrow 10 \rightarrow 6$. We collected the online request logs from January 13 to February 18, 2024. The collection process is as follows. We randomly sample 42K seed users on January 12, 2024, and store each recommendation request of the seed users from January 13, 2024. As shown in Figure 1, we sample some filtered videos from each stage but adopt a stage-wise strategy. From January 13 to February 04, 2024, which is called the 1st period, we sample 10 filtered videos of the pre-ranking stage named pre-rank_neg, 10 filtered videos of the coarse ranking stage named coarse_neg, top 10 ranking videos as rank_pos, 10 sampling filtered videos after the 120-th re-ranking video as rank_neg in the ranking stage, top 10 re-ranking videos as rerank_pos and 10 sampling filtered videos after the 80-th re-ranking video as rerank_neg in the re-ranking stage, and the user's various feedbacks on the exposed videos. Note that the recommendation scenario is feeds-style, the user can only watch one video on the screen. So, the 6 output videos of the RS may not all be exposed to the user because the user can leave the APP at any

Table 1: Detail quantity information of various aspects in RecFlow.

|  | #Stage Sample | #Request | #Users | #Realshow_videos | #All_videos |
|---|---|---|---|---|---|
| 1st Period | 352,120,401 | 6,062,348 | 38,193 | 5,984,924 | 30,305,725 |
| 2nd Period | 1,572,217,303 | 3,308,233 | 35,073 | 3,627,694 | 55,665,503 |
| Total | 1,924,337,704 | 9,370,581 | 42,472 | 8,773,147 | 82,216,301 |
|  | #Realshow | #Like | #Long_view | #Effective_view | #Follow |
| 1st Period | 24,523,473 | 1,027,013 | 5,853,054 | 9,343,776 | 69,495 |
| 2nd Period | 13,721,842 | 618,158 | 3,111,439 | 5,063,751 | 37,558 |
| Total | 38,245,315 | 1,645,171 | 8,964,493 | 14,407,527 | 107,053 |
|  | #Forward | #Comment | #Prerank_neg | #coarse_neg | #Rank_pos |
| 1st Period | 45,966 | 175,896 | 60,623,480 | 60,623,480 | 60,624,430 |
| 2nd Period | 23,769 | 114,741 | 132,329,320 | 132,329,320 | 33,082,330 |
| Total | 69,735 | 290,637 | 192,952,800 | 192,952,800 | 93,706,760 |
|  | #Rank_neg | #Rank | #Rerank_pos | #Rerank_neg | #Re-rank |
| 1st Period | 60,624,012 | 121,248,442 | 60,624,613 | 60,623,606 | 121,248,219 |
| 2nd Period | 33,082,330 | 1,307,558,663 | 33,082,330 | 33,082,330 | 1,307,558,663 |
| Total | 93,706,342 | 1,428,807,105 | 93,706,943 | 93,705,936 | 1,428,806,882 |

time. We define the realshow field to identify whether the user has watched the video. From February 05 to February 18, 2024, which is called the 2nd period, we expand the amount of stage samples. Both the pre-ranking_neg and the coarse_neg go up to 40. For the ranking, re-ranking, and edge ranking stages, we save all the videos that appear in these stages. We still obtain the rank_pos, rank_neg, rerank_pos, rerank_neg, and realshow under the same stage-wise strategy as the previous period. We collect stage samples in this way, considering the storage pressure and information integrity. The 2nd period has more complete stage information compared to the 1st period, which gives the researchers more choices to further process the dataset based on their needs. We sample 10/40 filtered videos from the pre-ranking and coarse ranking stages because keeping all of the filtered videos has huge storage pressure. Besides, the videos filtered by the first three stages are less important. For the latter three stages, we keep the information integrity of the stage. The videos appearing in these stages are closer to the user's preference and have a small scale.

## 2.2 FEATURES

The formation of each instance in RecFlow is {*request_id, request_timestamp, user_id, device_id, age, gender, province, video_id, author_id, category_level_one, category_level_two, upload_type, upload_timestamp, duration, realshow, rerank_pos, rerank_neg, rank_pos, rank_neg, coarse_neg, pre-rank_neg, rank_index, rerank_index, playing_time, effective_view, long_view, like, follow, forward, comment*}. *realshow* indicates whether the user has watched the video. The same procedure is applied to the other *_pos/neg* fields. For example, when the video ranks top 10 in the ranking stage, then the *rank_pos* is set to 1 otherwise 0. To reserve the original industrial RS information, we also retain the ranking position of each video in the ranking and reranking stages through the *rank_index* and *rerank_index* fields. We record seven types of positive feedback that reflect the user's varying degrees of preference towards videos. *playing_time* is the time the user spends watching the video. The other features' details are in the subsection Feature Description A.1 of Appendix.

## 2.3 ANALYSIS

In this section, we conduct a basic statistical analysis to show RecFlow's characteristics. We collect 9 million requests. It has 38 million exposure samples and 1.9 billion stage samples (including exposure samples). Among these samples, there are 42K users, 8.7 million exposed videos, and 82 million videos. Nearly 89% of videos are not exposed. This new character does not exist in existing RS datasets. During the first period, the quantity of each defined stage's samples is about 60 million. Stage samples are 14.8x larger than exposed samples. The difference between stage samples and exposure samples has increased to 236 times in the 2nd period. The huge quantity difference is the foundation for studying the distribution shift problem. The detailed quantities of

the dataset are shown in Table 1. Figure 3, whose horizontal axis represents the range of the number of videos interacted with by users and the vertical axis shows the number and percentage of users within that range, illustrates that the frequency of users exhibits a long-tail distribution. In Figure 4, the horizontal axis represents the logarithm of the frequency of video appearances, while the vertical axis shows the video quantity corresponding to that frequency. The left chart only includes videos marked as *realshow* with 1, which are the exposed videos, while the right chart includes videos from all stages. It shows the frequency of videos in exposure space and all stages' space, respectively. The left chart shows that exposure video frequency follows a long-tail distribution. The right chart reveals that video frequency in all stages also obeys the long-tail distribution, which is new discovery.

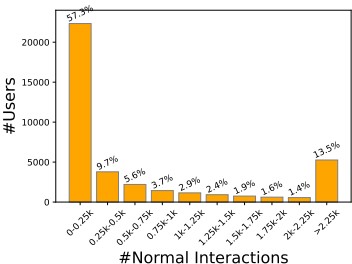

Figure 3: User Distribution.

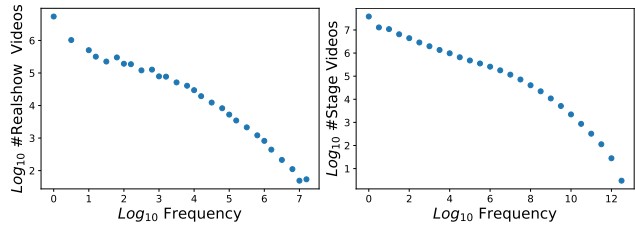

Figure 4: Video Distribution.

## 2.4 COMPARISON

We compare RecFlow with existing RS datasets. MovieLens (Harper & Konstan, 2015) contains the user's rating data for movies. Amazon (Ni et al., 2019) dataset contains the user's review information on the product. Yelp (Asghar, 2016) is a dataset for location recommendation. The three datasets only contain the user's single type of positive feedback. Taobao (Zhu et al., 2018), an e-commerce dataset, has four types of the user's positive feedback. Tenrec (Yuan et al., 2022) is a comprehensive recommendation dataset that captures multiple types of user feedback across four distinct recommendation scenarios. KuaiRec (Gao et al., 2022a) is a full-observed video recommedation dataset. KuaiRand (Gao et al., 2022b) is an unbiased sequential video recommendation dataset with randomly exposed videos. KuaiSAR (Sun et al., 2023) is a unified search and recommendation dataset. The three datasets are opened for dedicated research problems. RecFlow differs from those datasets because of the existence of samples from each recommendation stage. Table 8 in the subsection Dataset Comparison A.2 of Appendix gives a detailed comparison between RecFlow and existing RS datasets. We also discuss the limitation of RecFlow in subsection A.3 of the Appendix.

## 2.5 USER CONSENT AND PRIVACY PROTECTION

We only collect interaction data from the user who has made his/her personal information publicly (like user_id, age, gender, province, etc), and this public information allows for some level of data sharing, according to the privacy policy that users voluntarily agreed to when they signed up for an account. Besides, we anonymize all features that contain personal information. In detail, we anonymize each feature ID by adding the raw ID value with a random large integer first and remapping it to a new ID through the Hash algorithm. It can not know who the person in the real world is from the anonymous data. The General Data Protection Regulation of the European Union has confirmed that "personal information that has been anonymized does not belong to personal information. Therefore, personal information that has been anonymized does not have the corresponding personal information compliance obligations, and companies can freely process it without the consent of individuals." Thus, our open-source dataset meets legal requirements.

We have anonymized all features which contain personal information including request_id, user_id, device_id, age, gender, province, video_id, author_id, category_level_one, category_level_two, and upload_type. We first anonymize each feature ID by adding the raw ID value with a random large integer and then remapping it to a new ID through the Hash algorithm. Note that each raw ID value owns a unique larger integer. The rest of the features are stage labels and the user's feedback labels, which are not related to privacy. Anonymizing data with random noise and the Hash algorithm

satisfies the privacy protection requirements of the law of the European Union. The way of RecFlow's anonymization is more strict than previous public recommendation datasets, including Amazon (Ni et al., 2019), Taobao (Zhu et al., 2018), KuaiRec (Gao et al., 2022a), and Tenrec (Yuan et al., 2022). We add random large integer noise before the Hash algorithm and others not. It is nearly impossible to recover raw personal information, such as who the person in the real world is.

# 3 EXPERIMENTS

We explore how to utilize stage samples to alleviate distribution shift and distill knowledge of subsequent stages for improving RS's performance. We focus on typical retrieval, coarse ranking, and ranking stages. For each stage, we briefly introduce its duty and existing learning paradigm. Then, we state the motivation and the ways of exploiting stage samples. Finally, we report the experiment results and analysis. We run all experiments five times with Pytorch (Imambi et al., 2021) on Nvidia 32G V100. We report the average result and standard deviation. For all methods and all experiments, we train the neural models for only one epoch, and there is no early stopping. Thus, all methods are compared fairly. There are two reasons for only one epoch. First, all online recommendation models of the industrial RS are trained by one epoch. We keep consistency with the online configuration. Second, there exists a one-epoch phenomenon (Zhang et al., 2022) of the training recommendation model, which indicates that multi-epoch training does not bring improvement.

## 3.1 RETRIEVAL

Retrieval is the first stage of the industrial RS. It aims at retrieving thousands of videos that the user potentially prefers from the 100 million scale video corpus. Given the large candidate pool, the retrieval stage mostly adopts the lightweight two-tower model together with approximate nearest neighbor search to retrieve items quickly. To ensure that the user's preferred videos are obtained, the retrieval models usually learn with positive feedback videos as positive samples and randomly sampling videos as negative samples. We choose SASRec (Kang & McAuley, 2018) with one head and one layer for exploration experiments. We apply the effective_view videos as positive samples and randomly sample 200 videos as negative samples for each positive. To keep consistency with the real industrial RS's online learning mode, we train SASRec with the first 36 days' data day by day. The data from the last day is for evaluation. We utilize the standard top-N ranking metrics, including hit Recall@K and NDCG@K. K is set to 100, 500, 1000. The feature is the user's 50 past effective_view videos. We apply embedding for the *video_id* feature and set the embedding dimension to 8. The batch size is $4,096$ and the learning rate is $1e-1$. BPR (Rendle et al., 2012) is the loss function, and Adam (Kingma & Ba, 2014) is used for optimization.

### 3.1.1 HARD NEGATIVE MINING

Recent research (Zhang et al., 2013; Rendle & Freudenthaler, 2014; Lian et al., 2020) has shown that hard negative mining usually not only accelerates the convergence but also improves the model accuracy for the retrieval model. The hard negative samples are those videos that are similar to the positive videos but uninteresting to the user. The multi-stage RS pipeline aims at estimating the user's preference. Videos that fail to be exposed to the user during the pipeline are similar to the displayed positive video but very likely less attractive to the user. Thus, we think the unexposed stage samples indeed satisfy the definition of hard negative samples. We conduct experiments to explore the effectiveness of the stage samples as hard negative samples. In the experiments, we replace some randomly sampled easy negative samples with the same number of hard negative stage samples. The total number of negative videos for each positive video is 200.

We have the following findings from the result in Table 2. (1) Applying filtered videos from each stage as hard negatives all gains performance improvement on the Recall/NDCG metric. (2) As the K in Recall/NDCG@K becomes smaller, the performance improvement becomes better. For example, when we add 1 *pre-rank_neg* as hard negative, the relative promotion of Recall@100, 500, 1000 are 24.7%, 18.2%, 9.2% respectively, and the relative promotion of NDCG@100, 500, 1000 are 28.3%, 20.7%, 12.6% respectively. (3) The hard negative video from *rerank_pos* outperforms than the other stages. We think that videos from *rerank_pos* are negative samples of appropriate difficulty. We also vary the number of hard negative samples to observe the changes in effectiveness. The experiment result and analysis are in the subsection A.4 of the Appendix.

Table 2: Recall(R) and NDCG(N) results (mean $\pm$ std) obtained by using a single different stage sample as the hard negative sample during the retrieval stage, with units of %. The best and baseline results are based on the paired $t$-test at the significance level 5%.

| Hard Negative Type | R@100 | N@100 | R@500 | N@500 | R@1000 | N@1000 |
|---|---|---|---|---|---|---|
| Baseline | 0.461±0.085 | 0.099±0.085 | 1.593±0.229 | 0.241±0.045 | 2.685±0.186 | 0.356±0.040 |
| Prerank_neg | 0.575±0.095 | 0.127±0.028 | 1.883±0.170 | 0.291±0.030 | **2.931±0.142** | 0.401±0.030 |
| Coarse_neg | 0.555±0.066 | 0.121±0.021 | 1.729±0.152 | 0.267±0.033 | 2.758±0.169 | 0.376±0.035 |
| Rank_neg | 0.462±0.126 | 0.094±0.030 | 1.695±0.230 | 0.249±0.043 | 2.733±0.221 | 0.359±0.042 |
| Rank_pos | 0.648±0.074 | 0.134±0.017 | 1.794±0.187 | 0.277±0.028 | 2.737±0.173 | 0.376±0.025 |
| Rerank_neg | 0.577±0.091 | 0.119±0.019 | 1.804±0.208 | 0.274±0.034 | 2.724±0.242 | 0.371±0.036 |
| Rerank_pos | **0.687±0.087** | **0.144±0.018** | **1.889±0.108** | **0.295±0.021** | 2.892±0.105 | 0.401±0.020 |
| Exposure_neg | 0.603±0.093 | 0.137±0.016 | 1.860±0.207 | **0.295±0.032** | 2.902±0.221 | **0.405±0.033** |

Table 3: Recall(R) and NDCG(N) results (mean $\pm$ std) obtained by using a single different stage sample as the cascade sample during the retrieval stage, with units of %. The best and baseline results are based on the paired $t$-test at the significance level 5%.

| Cascade Type | R@100 | N@100 | R@500 | N@500 | R@1000 | N@1000 |
|---|---|---|---|---|---|---|
| Baseline | 0.461±0.085 | 0.099±0.085 | 1.593±0.229 | 0.241±0.045 | 2.685±0.186 | 0.356±0.040 |
| Prerank_neg | 0.677±0.061 | 0.167±0.041 | 2.268±0.129 | 0.367±0.048 | 3.446±0.111 | 0.492±0.042 |
| Coarse_neg | 0.665±0.120 | 0.163±0.045 | 2.253±0.052 | 0.361±0.037 | 3.371±0.090 | 0.479±0.038 |
| Rank_neg | 0.704±0.150 | 0.173±0.049 | 2.282±0.250 | 0.373±0.055 | 3.410±0.203 | 0.491±0.052 |
| Rank_pos | 0.685±0.094 | 0.151±0.025 | 2.191±0.085 | 0.340±0.023 | 3.346±0.078 | 0.462±0.019 |
| Rerank_neg | 0.707±0.083 | 0.163±0.024 | 2.273±0.121 | 0.359±0.024 | 3.338±0.083 | 0.471±0.022 |
| Rerank_pos | 0.795±0.108 | 0.176±0.025 | 2.263±0.078 | 0.361±0.017 | 3.394±0.048 | 0.480±0.016 |
| Exposure_neg | 0.692±0.071 | 0.156±0.028 | 2.150±0.108 | 0.340±0.033 | 3.266±0.183 | 0.458±0.036 |
| FS-LTR | **0.803±0.095** | **0.215±0.027** | **2.466±0.090** | **0.425±0.029** | **3.606±0.060** | **0.545±0.024** |

### 3.1.2 INTERPLAY BETWEEN RETRIEVAL AND SUBSEQUENT STAGES

The most important characteristic of industrial RS is the multi-stage. Every stage has its duty and mature paradigm. The goal of each stage is consistent, which is to fit the user's preference. Although models of all stages aim at fitting the user's preference, they can not capture the user's preference perfectly. Few people focus on the interplay between stages. The academic researchers lack available datasets and the industrial engineers only devote effort to the assigned stage. (Zheng et al., 2024) has pointed out that there are two factors influencing the video's exposure and the user's feedback. First, it is the user's preference on the video. Second, it is the preference of the subsequent stage towards the video. For example, one video that the user likes is retrieved during the retrieving stage but is filtered out by the ranking model due to its imperfect preference estimation ability. This video is inefficient for the whole RS because it can not be exposed to the user at all. The optimal solution for the model of each stage is to select videos that satisfy the preference of the user and subsequent stages simultaneously. FS-LTR (Zheng et al., 2024) has proposed the Generalized Probability Ranking Principle (GPRP) to prove that the solution proposed above is optimal theoretically. We implement FS-LTR in this section to see its effectiveness. The user's preference can be learned from the positive feedback samples and randomly sampled negative samples. To learn the preference of subsequent stages, we introduce additional ranking loss, which forces the logits of samples from high-priority stages to be bigger than the logits of samples from low-priority stages. The priority of stages are {positive:6, exposure_neg:5, rerank_pos:4, rank_pos:4, rerank_neg:3, rank_neg:3, corase_neg:2, pre-rank_neg:1, random_neg:0}. Exposure_neg represents the video that has been exposed to the user (realshow=1) but obtains negative feedback. This definition of priority applies throughout the paper. We always keep one positive sample with 200 negative samples. We first introduce the stage preference one stage at a time by replacing random negatives with stage samples with BPR as Eq( 1):

$$L_{FS\text{-}LTR} = \sum_{i=1}^{N} \sum_{j \in \{k:p_k < p_i\}} BPR(o_i, o_j) \tag{1}$$

where $N$ equals 200, $p_{i(k)}$ represents the priority level of sample $i(k)$, and $p_k < p_i$ means the priority level of sample $k$ is lower than sample $i$.

We have the following findings from experiment results in Table 3. (1) FS-LTR gains performance enhancement when introducing the sample of each stage respectively compared to the baseline. (2) Under the same negative setting, FS-LTR can achieve better results compared with the results of hard negative mining in Table 2. (3) As the K of Recall@K becomes smaller, the performance improvement becomes better. For example, when we add 1 *pre-rank_neg* as hard negative, the relative promotion of Recall@100, 500, 1000 are $46.8\%, 42.4\%, 28.3\%$ respectively. NDCG@K holds the same trends. We also try to introduce multiple samples from more stages gradually to investigate the effectiveness of modeling more subsequent stages' preferences in subsection A.5 of the Appendix.

## 3.2 COARSE RANKING

Coarse ranking receives favorable videos from the pre-ranking stage and filters less favorable videos to fulfill its duty. As the candidate videos are more similar and not easy to distinguish, coarse ranking models take more feature fields as input and use a more complex neural network to ensure their modeling capacity. However, there are $3,000$ videos to be scored in our scenario, the two-tower structure is still the best choice. We take DSSM (Huang et al., 2013) as the coarse ranking model. The Multi-layer Perceptron (MLP) of the user and video towers in DSSM are set to be [128, 64, 32]. Existing coarse ranking models are almost learned through the exposure of positive and negative samples. AUC (Area Under the Curve) on the testing exposure samples is employed to assess the algorithm's performance. *Effective_view* is the learning signal. Following the retrieval experiment, data from the first 36 days is for training and the last day's data is for evaluation. The feature fields include *user_id, device_id, age, gender, province, video_id, author_id, category_level_one, category_level_two, upload_type, upload_timestamp, request_timestamp*. The *upload_timestamp* and *request_timestamp* are divided into the *week, day, hour* feature fields. Besides, we add the user's past 50 effective_videos as the behavior sequence. We process effective behavior sequences through mean pooling. We apply embedding for all the feature fields and set the embedding dimension to $8$. The batch size is $1,024$ and the learning rate is $1\mathrm{e}{-2}$. Binary Cross Entropy is the loss, and Adam is used for optimization. We also utilize the stage samples to explore the auxiliary ranking task and user behavior sequence modeling in the coarse ranking model, both of which boost the AUC metric greatly. The methods, together with the experiment results and analysis, are in subsections A.6 and A.7 of the Appendix.

### 3.2.1 DATA DISTRIBUTION SHIFT

Data distribution shift is a longstanding problem in RS. Due to the absence of datasets containing stage samples, few works (Ma et al., 2018b; Qin et al., 2022) focus on the problem in the coarse ranking stage. The coarse ranking model is trained based on the exposed samples which contains 6 videos at most but has to score $3,000$ videos in each request. The data distribution between training and testing exists a huge inconsistency. What's worse, the AUC metric evaluated on the exposure space for guiding the offline algorithm's optimization is inconsistent with the online scenario (Song et al., 2022; Zhang et al., 2023b). The collected stage samples make the evaluation space more consistent with the online situation. Following (Zhang et al., 2023b), we apply the Recall@K metric, which is consistent with the effect of online business. Because we saved all the videos in the ranking stage on February 18, 2024, the candidate set for calculating the Recall@K and NDCG@K is composed of the videos in the ranking stage together with videos of coarse_neg. We set K to $100, 200$. We also report the classical AUC metric. We try to directly supplement the stage samples as extra negative samples into the training data. Although it's possible to introduce false negative videos, this can still largely reduce the difference in data distribution between training and testing. However, supplementing extra negative samples increases the machine overload. Thus, we show the relationship between the performance and the quantity of the additional negative samples.

We have the following conclusions from the result in Table 4. (1) Supplementing stage videos as extra negative samples can largely enhance the Recall and NDCG metric. The improvement can be attributed to the consistency of data distribution between training and testing. (2) When increasing the quantity of extra negative samples, the improvement becomes greater. And introducing partial or all videos from all corresponding stages gains the best results. This indicates that the more consistent the data distribution between training and testing, the more improvement. (3) Introducing videos from *rank_pos* and *rerank_pos* gains light enhancement compared to *coarse/rank/rerank_neg*. We think that there are false negative samples that mislead the model's learning. (4) The classical AUC metric has

Table 4: The result (mean $\pm$ std) of using different stages' samples as extra negatives for Coarse Ranking. The best and baseline results are based on the paired $t$-test at the significance level $5\%$.

| Neg Type | #N | AUC | LogLoss | Recall@100 | NDCG@100 | Recall@200 | NDCG@200 |
|---|---|---|---|---|---|---|---|
| Baseline | - | **0.718±0.001** | **0.592±0.003** | 0.271±0.027 | 0.059±0.027 | 0.535±0.009 | 0.096±0.003 |
| Coarse_neg | 1 | 0.705±0.002 | 0.608±0.006 | 0.321±0.012 | 0.072±0.027 | 0.597±0.038 | 0.111±0.003 |
| | 10 | 0.633±0.016 | 0.773±0.018 | 0.392±0.012 | 0.088±0.004 | 0.668±0.007 | 0.126±0.003 |
| Rank_neg | 1 | 0.704±0.002 | 0.615±0.004 | 0.353±0.013 | 0.079±0.003 | 0.638±0.004 | 0.118±0.002 |
| | 10 | 0.618±0.016 | 0.825±0.027 | 0.454±0.011 | 0.102±0.005 | 0.726±0.005 | 0.140±0.004 |
| Rank_pos | 1 | 0.704±0.001 | 0.603±0.001 | 0.275±0.027 | 0.061±0.004 | 0.557±0.005 | 0.100±0.001 |
| | 10 | 0.623±0.020 | 0.769±0.003 | 0.290±0.002 | 0.069±0.004 | 0.591±0.019 | 0.111±0.004 |
| Rerank_neg | 1 | 0.702±0.001 | 0.616±0.005 | 0.337±0.007 | 0.076±0.001 | 0.605±0.002 | 0.113±0.001 |
| | 10 | 0.608±0.021 | 0.821±0.019 | 0.380±0.014 | 0.084±0.003 | 0.673±0.004 | 0.125±0.003 |
| Rerank_pos | 1 | 0.703±0.001 | 0.607±0.003 | 0.264±0.011 | 0.060±0.003 | 0.548±0.003 | 0.099±0.002 |
| | 10 | 0.618±0.024 | 0.782±0.025 | 0.285±0.015 | 0.069±0.003 | 0.587±0.011 | 0.111±0.003 |
| All | 1 | 0.662±0.006 | 0.704±0.011 | 0.386±0.010 | 0.084±0.001 | 0.676±0.008 | 0.125±0.001 |
| | 10 | 0.563±0.004 | 1.243±0.030 | **0.455±0.004** | **0.105±0.001** | **0.728±0.004** | **0.144±0.001** |

Table 5: The result (mean $\pm$ std) of interplay between Coarse Ranking and Subsequent Stages. The best and baseline results are based on the paired $t$-test at the significance level $5\%$.

| Method | AUC | LogLoss | Recall@100 | NDCG@100 | Recall@200 | NDCG@200 |
|---|---|---|---|---|---|---|
| Baseline | **0.718±0.001** | **0.592±0.003** | 0.271±0.027 | 0.059±0.027 | 0.535±0.009 | 0.096±(0.003 |
| PositiveRank | 0.554±0.005 | 1.040±0.051 | 0.457±0.001 | 0.112±0.001 | 0.723±0.002 | 0.149±0.001 |
| FS-LTR | 0.473±0.013 | 1.253±0.071 | **0.475±0.002** | **0.119±0.001** | **0.734±0.002** | **0.155±0.001** |

opposite trends. After adding extra negative samples, the gap between the training data distribution and the data distribution of exposure space for evaluating AUC enlarges. As we mentioned in the retrieval section, there exists a hardness level among different stage samples. Expanding negatives degrades the model's ability to distinguish hard negatives (exposed un-effective_view samples) but enhances the capability of recognizing less hard negatives (videos from stages).

### 3.2.2 INTERPLAY BETWEEN COARSE RANKING AND SUBSEQUENT STAGES

FS-LTR is a general principle and is applicable in the coarse ranking stage. We implement FS-LTR with samples of *positive, exposure_neg, rerank_pos, rerank_neg, rank_pos, rank_neg, coarse_neg*, which is the inference space of the coarse ranking model. In order to apply the loss 1, we aggregate samples of the same request into the same batch. We also add a contrast experiment, PostiveRank, in which we just make the logits of positive samples bigger than the logits of the other samples. The result in Table 5 shows that FS-LTR can achieve the best performance on the Recall/NDCG. It demonstrates the necessity of learning the preferences of both the user and the subsequent stages.

### 3.3 RANKING

Ranking is nearly the most important stage in the industrial multi-stage RS and has been studied sufficiently. It determines the displayed items to the user. Its candidate video set is the output of the coarse ranking stage. Given the importance and difficulty of the task, ranking model has the most complex neural network structure and uses most feature fields. The time cost is acceptable because it only needs to score 500 videos. We utilize DIN (Zhou et al., 2018) as the ranking model. The architecture of the DIN's MLP is [128, 128, 32, 1]. Ranking model is also learned on the exposure space and evaluates AUC on testing exposure samples. For the experiment settings, the ranking model remains the same as the coarse ranking model. We also use the stage samples to explore the auxiliary ranking task and user behavior sequence modeling in the ranking model, both of which improve the classical AUC greatly. Detail of methods and experiments are in subsection A.8 and A.9 of the Appendix.

Table 6: The result (mean $\pm$ std) of using different stages' samples as extra negatives for Ranking. The best and baseline results are based on the paired $t$-test at the significance level 5%.

| Neg Type | #N | AUC | LogLoss | Recall@50 | NDCG@50 | Recall@100 | NDCG@100 |
|---|---|---|---|---|---|---|---|
| Baseline | - | **0.727±0.001** | **0.583±0.003** | 0.169±0.005 | 0.045±0.002 | 0.319±0.008 | 0.069±0.002 |
| Rank_neg | 1 | 0.711±0.001 | 0.610±0.008 | 0.223±0.005 | 0.061±0.002 | 0.395±0.007 | 0.088±0.002 |
|  | 10 | 0.645±0.003 | 0.810±0.032 | 0.264± 0.012 | 0.074±0.004 | 0.454±0.014 | 0.105±0.005 |
| Rank_pos | 1 | 0.711±0.001 | 0.604±0.008 | 0.176±0.005 | 0.047±0.001 | 0.327±0.010 | 0.072±0.002 |
|  | 10 | 0.653±0.002 | 0.724±0.029 | 0.185±0.009 | 0.049±0.003 | 0.331±0.015 | 0.073±0.003 |
| Rerank_neg | 1 | 0.708±0.001 | 0.616±0.006 | 0.215±0.003 | 0.059±0.001 | 0.380±0.006 | 0.085±0.001 |
|  | 10 | 0.624±0.005 | 0.815±0.028 | 0.232±0.018 | 0.064±0.005 | 0.406±0.031 | 0.092±0.007 |
| Rerank_pos | 1 | 0.711±0.002 | 0.608±0.006 | 0.170±0.012 | 0.045±0.003 | 0.319±0.019 | 0.069±0.004 |
|  | 10 | 0.646±0.002 | 0.782±0.033 | 0.183±0.016 | 0.048±0.005 | 0.335±0.009 | 0.073±0.003 |
| All | 1 | 0.675±0.003 | 0.697±0.010 | 0.234±0.005 | 0.064±0.002 | 0.411±0.004 | 0.093±0.002 |
|  | 10 | 0.602±0.005 | 1.076±0.049 | **0.278±0.027** | **0.078±0.007** | **0.467±0.038** | **0.108±0.009** |

Table 7: The result (mean $\pm$ std) of interplay between Ranking and Subsequent Stages. The best and baseline results are based on the paired $t$-test at the significance level 5%.

| Method | AUC | LogLoss | R@50 | N@50 | R@100 | N@100 |
|---|---|---|---|---|---|---|
| Baseline | **0.727±0.001** | **0.583±0.003** | 0.169±0.005 | 0.045±0.002 | 0.319±0.008 | 0.069±0.002 |
| PositiveRank | 0.564±0.003 | 1.466±0.313 | 0.309±0.016 | 0.093±0.006 | 0.506±0.014 | 0.125±0.006 |
| FS-LTR | 0.461±0.005 | 1.215±0.391 | **0.323±0.012** | **0.098±0.003** | **0.525±0.014** | **0.131±0.004** |

### 3.3.1 DATA DISTRIBUTION SHIFT

The ranking model also suffers from the data distribution shift problem. In each request, there are at most 6 exposure samples for training but 500 videos to be scored. The data distribution gap between training and testing still exists. Fortunately, the inconsistency is not as serious as the coarse ranking model. The exploration experiment setting for alleviating the data distribution shift problem is the same as the coarse ranking model, including motivation, method, and evaluation metrics. The difference is that samples of *coarse_neg* are excluded from training and evaluation because they are not in the ranking model's candidate video set. The result is shown in Table 6. We can find that the more consistent the data distribution between training and testing, the more the Recall and NDCG improve. Other conclusions are the same as coarse ranking, and we don't repeat them here.

### 3.3.2 INTERPLAY BETWEEN RANKING AND SUBSEQUENT STAGES

We also conduct FS-LTR in the ranking stage. The experiment settings are mostly the same as coarse ranking except that training samples are from *positive, exposure_neg, rank_neg, rank_pos, rerank_neg, rerank_pos*, which is the inference space of the ranking model. PositiveRank serves as the contrast purpose. The results in Table 7 lead to the same conclusions as coarse ranking.

## 4 CONCLUSIONS

In this paper, we propose a new dataset called RecFlow which captures information across the entire pipeline of an industrial recommendation system. It will provide researchers with convenience for studying multi-stage recommendations. We also conduct extensive preliminary experiments using RecFlow in retrieval, coarse ranking, and ranking stages. The experimental results demonstrate that utilizing stage samples indeed enhances recommendations.

ACKNOWLEDGEMENTS

The work was supported by grants from the National Natural Science Foundation of China (No. U24A20253).

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

# A APPENDIX

## A.1 FEATURE DESCRIPTION

The *request_id* identifies each recommendation request and *request_timestamp* represents the time when the recommendation request arises. Every user has a unique ID named *user_id*. *device_id* means the device that initiates the recommendation request. We also provide the user's profile information including *age, gender, province*. *Age* is grouped into ten buckets. *video_id* identifies each video. *author_id* represents the one who uploads the video. We also record the video's attributes involving *category_level_one, category_level_two, upload_type, upload_timestamp, duration*. *category_level_one, category_level_two* are categories of the video, where *category_level_one* is the coarse-grained category (e.g., sports, history, K-pop, etc.) and *category_level_two* indicates the fine-grained category (e.g., UEFA Champions League, Ming Dynasty, BLACKPINK, etc). The *upload_type* and *upload_timestamp* stand for the type of the video (e.g., micro-video, long-video, picture, etc) and the time when the video was uploaded. *duration* is the video's lasting time. Next, we describe the fields identifying the stage information. The *effective_view* and *long_view* are the binary features (0 and 1) defined according to business interest. *long_view* is more strict than *effective_view*. *like* indicates whether the user clicks the ♡ button. *follow* means the user follows the video's author. *forward* represents the user sharing the video. *comment* stands for whether the user makes some text review about the video. Note that the feedback values of the unexposed video are all set to 0. The fields of *request_id, user_id, device_id, age, gender, province, video_id, author_id, category_level_one, category_level_two, upload_type* are all have been anonymized ensuring the privacy protection.

## A.2 DATASET COMPARISON

Table 8: The characteristic comparison of different recommendation datasets.

| Dataset | Stage Sample | Type_feedbacks | #Users | #Interactions | True_neg | Req_id |
|---|---|---|---|---|---|---|
| MovieLens-20M (Harper & Konstan, 2015) | ✗ | 1 | 138K | 20M | ✗ | ✗ |
| Amazon (Ni et al., 2019) | ✗ | 2 | / | 233M | ✗ | ✗ |
| Yelp (Asghar, 2016) | ✗ | 1 | 1.9M | 8M | ✗ | ✓ |
| Taobao (Zhu et al., 2018) | ✗ | 4 | 987K | 100M | ✗ | ✗ |
| TenRec-QKV (Yuan et al., 2022) | ✗ | 4 | 5.0M | 142M | ✓ | ✗ |
| TenRec-QKA (Yuan et al., 2022) | ✗ | 6 | 1.3M | 46M | ✗ | ✗ |
| KuaiRec (Gao et al., 2022a) | ✗ | 1 | 7K | 12M | ✓ | ✗ |
| KuaiRand (Gao et al., 2022b) | ✗ | 6 | 27K | 322M | ✓ | ✗ |
| KuaiSAR (Sun et al., 2023) | ✗ | 9 | 26K | 19M | ✓ | ✗ |
| RecFlow | ✓(1.9B) | 7 | 42K | 38M | ✓ | ✓ |

## A.3 LIMITATIONS

RecFlow, while valuable in lots of recommendation research problems, also has its own drawbacks. Understanding advantages and disadvantages is vital for ensuring accurate academic use. First, we collect data from only one recommendation scenario which causes RecFlow can not be applied to the multi/cross-domain recommendation. Second, RecFlow can't advance the research of multimodal recommendation because of lacking multimodal features such as text and image. On the other hand, it needs more hardware resource cost because RecFlow contains 1,924,337,704 instances.

## A.4 RETRIEVAL: THE EFFECT OF THE NUMBER OF HARD NEGATIVES IN RETRIEVAL STAGE

The result in Table 9 is the result of varying the number of hard negative samples from each stage. We set the number to 2 and 10 for observation. (1) Increasing the number of hard negative videos from *prerank_neg* can further improve the performance but with diminishing marginal effect. (2) For *coarse_neg, rank_pos, rerank_neg, rerank_pos, exposure_neg*, adding videos from them as hard negative samples degrades the performance. The closer the stage to the positive feedback, the more

Table 9: Recall(R) and NDCG(N) results obtained by using 2 or 10 different stage samples as the hard negative sample during the retrieval stage, with units of %.

| HN Type | #HN | R@100 | N@100 | R@500 | N@500 | R@1000 | N@1000 |
|---|---|---|---|---|---|---|---|
| Baseline | - | 0.461 | 0.099 | 1.593 | 0.241 | 2.685 | 0.356 |
| Prerank_neg | 2 | 0.474 | 0.108 | 1.664 | 0.257 | 2.574 | 0.352 |
| | 10 | 0.457 | 0.101 | 1.515 | 0.236 | 2.319 | 0.321 |
| Coarse_neg | 2 | **0.634** | 0.140 | **1.948** | **0.305** | 2.844 | 0.400 |
| | 10 | 0.524 | 0.125 | 1.564 | 0.256 | 2.347 | 0.339 |
| Rank_neg | 2 | 0.492 | 0.104 | 1.687 | 0.254 | 2.645 | 0.355 |
| | 10 | 0.323 | 0.069 | 1.321 | 0.193 | 2.231 | 0.289 |
| Rank_pos | 2 | 0.589 | 0.126 | 1.846 | 0.284 | 2.722 | 0.376 |
| | 10 | 0.544 | 0.109 | 1.544 | 0.235 | 2.310 | 0.315 |
| Rerank_neg | 2 | 0.606 | 0.120 | 1.849 | 0.277 | 2.831 | 0.381 |
| | 10 | 0.336 | 0.070 | 1.186 | 0.176 | 2.032 | 0.265 |
| Rerank_pos | 2 | 0.428 | 0.091 | 1.584 | 0.236 | 2.551 | 0.338 |
| | 10 | 0.219 | 0.043 | 0.954 | 0.135 | 1.819 | 0.226 |
| exposure_neg | 2 | 0.576 | 0.138 | 1.854 | 0.300 | **2.866** | **0.407** |
| | 10 | 0.629 | **0.142** | 1.924 | **0.305** | 2.856 | 0.403 |

Table 10: Recall(R) and NDCG(N) results obtained by using different combinations of stage samples as cascade samples during the retrieval stage, with units of %. CN-PN represents the use of coarse_neg and preran_neg. R-CN-PN represents the use of exposure_neg, coarse_neg, and prerank_neg. ALL represents the use of all stage samples in the current request.

| Cascade Type | R@100 | N@100 | R@500 | N@500 | R@1000 | N@1000 |
|---|---|---|---|---|---|---|
| Baseline | 0.461 | 0.099 | 1.593 | 0.241 | 2.685 | 0.356 |
| CN-PN | **0.803** | **0.215** | 2.466 | **0.425** | 3.606 | **0.545** |
| EN-CN-PN | 0.771 | 0.198 | **2.481** | 0.413 | **3.648** | 0.536 |
| All | 0.663 | 0.150 | 1.972 | 0.315 | 3.018 | 0.425 |

degradation. The phenomenon demonstrates that there exists a hardness level between videos from different stages. The closer to the positive feedback the stage, the more difficult for the retrieval model to distinguish. (3) When we increase the number of videos from *rank_neg* from 1 to 2, the performance is somewhat boosted. However, it still suffers from a severe performance drop when taking 10 *rank_neg* hard negative videos. As pointed out in (He et al., 2014; Zhang et al., 2023a), the ratio between easy and hard negatives has a critical influence on performance. We guess that harder negatives need more easy negatives and leave the hardness and ratio of hard negative samples for further research.

## A.5 RETRIEVAL: MORE RESULT OF RETRIEVAL'S INTERPLAY EXPERIMENT

The result of Table 10 is the experiment of introducing samples from more different stages into the FS-LTR. We can find that directly introducing samples from all stages is better than the baseline but is not the best. The setting of CN-PN achieves the best in our exploration, which indicates that ranking regularization between some priority levels may be unnecessary. We leave the in-depth exploration for future research.

## A.6 COARSE RANKING: AUXILIARY RANKING TASK

Increasing the ranking ability of the model trained with pointwise loss function (e.g., Click-through Rate prediction model) by adding an auxiliary ranking task has gained much attention recently (Yan et al., 2022; Bai et al., 2023; Sheng et al., 2023; Liu et al., 2024; Lin et al., 2024). The auxiliary

Table 11: The result of the auxiliary ranking task for the coarse ranking stage.

| Method | AUC | LogLoss | R@100 | N@100 | R@200 | N@200 |
|---|---|---|---|---|---|---|
| Baseline | 0.718 | 0.592 | 0.271 | 0.059 | 0.535 | **0.096** |
| w/ AuxLoss | **0.721** | **0.588** | **0.287** | **0.061** | **0.541** | **0.096** |

Table 12: The result of competitive relation modeling in UBM during coarse ranking stage.

| Method | AUC | LogLoss | R@100 | N@100 | R@200 | N@200 |
|---|---|---|---|---|---|---|
| Baseline | 0.718 | 0.592 | 0.271 | 0.059 | 0.535 | 0.096 |
| Competing Seq | **0.722** | **0.588** | **0.293** | **0.064** | **0.574** | **0.103** |

ranking task forces the logits of positive samples to be bigger than negative samples within the same batch or session through pairwise or listwise ranking loss. Inspired by these works, we propose a new auxiliary ranking task by forcing the logits of positive samples bigger than the stage samples of the same request. There is no ranking regularization on the negative samples. Note that the stage samples are only for auxiliary loss. The total loss function is as Eq( 2).

$$L = \frac{1}{N} \sum_{i=1}^{N} BCEWithLogit(o_i, y_i) + \alpha * \frac{1}{N_+ K} \sum_{j=1}^{N_+} \sum_{j_k=1}^{K} BPR(o_j, o_{j_k}) \qquad (2)$$

where $N$ is the batch size, $N_+$ is the number of positive samples in the batch, $j_k$ represents the stage sample within the same request as $j$, $K$ is the size of stage samples, and $o$ is the logit output by DSSM. $\alpha$ is the weight of auxiliary ranking loss. In the experiment, we use all the stage samples from *coarse_neg, rank_neg, rank_pos, rerank_neg, rerank_pos* stages. The result in Table 11 shows that the AUC increases by 0.002 and the Logloss decreases by 0.004, which is a significant improvement (Guo et al., 2017). Recall and NDCG also gain improvement. The designed auxiliary ranking task promotes both the classical and the newly proposed metrics, which demonstrates its effectiveness.

### A.7 COARSE RANKING: USER BEHAVIOR SEQUENCE MODELING

Competitive relation modeling has been attracting attention in user behavior sequence modeling (UBM) recently (Zheng et al., 2022; Hou et al., 2023; Fan et al., 2022; Li et al., 2023). Its motivation is that the user's feedback on items is also influenced by the displayed context. For example, if one user likes red T-shirts, he/she will click a pink T-shirt surrounded by items which he/she is not interested in, but he/she will click the red T-shirt surrounded by pink, blue, and yellow T-shirts. The competitive relation among displayed items has an impact on the user's feedback. There also exists a competitive relation among the videos in the *rerank/rank_pos* stages. These videos compete for exposure to the user. Inspired by (Hou et al., 2023), we explore introducing the competitive information in the stage samples into the UBM. For the user's past effective_view videos $S = [v_1, v_2, ..., v_{50}]$, we regard 10 videos in the *rank_pos* from the same request of each effective_view video in $S$ as the competitive information. We represent the competitive relation as $C = [[v_{1,1}, v_{1,2}, ..., v_{1,10}], [v_{2,1}, v_{2,2}, ..., v_{2,10}], ..., [v_{50,1}, v_{50,2}, ..., v_{50,10}]]$. We apply the hierarchical attention algorithm to model the competitive relation. First, we perform target attention between each effective_view behavior $v_i$ and its competing context $[v_{i,1}, v_{i,2}, ..., v_{i,10}]$. We will obtain the refined competing behavior representation $E = [c_1, c_2, ..., c_{50}]$. Then, we do mean pooling on $E$ to the user's competitive relation aware interest $competing\_interest$. Table 12 shows the experiment results. Both AUC and Logloss are improved significantly by 0.004. What's more, Recall@100,200 and NDCG@100,200 also get better performance. The video competitive information in the *rank_pos* is a useful signal for UBM. The result indicates that there exists a method that can improve both the classical AUC/Logloss and the newly applied Recall/NDCG metric.

### A.8 RANKING: AUXILIARY RANKING TASK

We also conduct the auxiliary ranking task in the ranking stage. The ranking loss is still Eq(2). The stage samples used in the ranking loss come from *rank_neg, rank_pos, rerank_neg, rerank_pos* stages.

Table 13: The result of the auxiliary ranking task for the ranking stage.

| Method | AUC | LogLoss | R@50 | N@50 | R@100 | N@100 |
|---|---|---|---|---|---|---|
| Baseline | 0.727 | **0.583** | **0.169** | **0.045** | **0.319** | **0.069** |
| w/ AuxLoss | **0.729** | **0.583** | 0.168 | **0.045** | 0.316 | 0.068 |

Table 14: The result of competitive relation modeling in UBM during ranking stage.

| | AUC | LogLoss | R@50 | N@50 | R@100 | N@100 |
|---|---|---|---|---|---|---|
| Baseline | 0.727 | 0.583 | **0.169** | **0.045** | 0.319 | **0.069** |
| Competing Seq | **0.732** | **0.578** | 0.168 | **0.045** | 0.313 | 0.068 |

The results are summarized in Table 13. We can draw findings the same with the auxiliary ranking task of coarse ranking, which verifies the broad effectiveness of the auxiliary ranking loss based on the stage samples.

## A.9 RANKING: USER BEHAVIOR SEQUENCE MODELING

Competing modeling is also explored in the ranking stage. We still choose the videos in the *rank_pos* as the competing context. We change the modeling method and make it more suitable for DIN. After acquiring the refined competing behavior sequence representation $E = [c_1, c_2, ..., c_{50}]$, we perform target attention between target video $v_{target}$ and $E$. Finally, we obtain the user's competing-aware interest $competing\_interest_t$ towards the target video $v_{target}$. The result in Table 14 shows that both the AUC and Logloss are improved by $0.005$ but the Recall and NDCG have no change. The result is not perfect as coarse ranking and it is worth exploring the modeling method continuously.

