# OpenReview forum: "RecFlow: An Industrial Full Flow Recommendation Dataset"
_ICLR.cc/2025/Conference — ICLR 2025 Poster_

### Official Review · Reviewer_Y4hS · 2024-10-28

**Soundness:** 3
**Presentation:** 3
**Contribution:** 3
**Rating:** 6
**Confidence:** 4

**Summary:**

This paper published a new dataset for multi-stage-funnel-based recommendation system, where the key difference from existing datasets is the inclusion of unexposed samples. Most existing datasets only contain samples that are exposed to users, and ranking and earlier-stage models are typically trained on exposed samples with user feedback. So inclusion of unexposed samples can facilitate research for many interesting problems in multi-stage recommendation, such as the distribution gap between training and serving, multi-stage consistency etc.

**Strengths:**

As far as I know, this is a first contribution of benchmark datasets that includes multi-stage and unexposed samples. Could be useful for researching important problems in multi-stage recommendation systems.

**Weaknesses:**

The evaluation criterion for the quality of a benchmark dataset for industrial recommendation system should be fidelity to an actual online recommendation system. For example, if researchers come up with new algorithms with metrics improvement using this dataset, then when it’s deployed to a real online system, such improvement can be validated. So it would be great if the authors can demonstrate such fidelity to some extent, e.g., by running online A/B test to compare the online performance and offline metrics to see the correlation.

The rules for determining how many samples for each stage seem quite ad-hoc w/o explanation of considerations. Could you explain the considerations that went into determining the number of samples collected at each stage? Are these numbers representative of typical production systems?

some typos:
line 239: datatse -> dataset
line 256: quote in wrong direction
line 296/399: 1e-1/1e-2 not well formatted
line 308/361: randomly sampling -> randomly sampled

**Questions:**

what’s the rationale for partitioning the data collection into two periods? Understanding the rationale for partitioning the data collection into two periods would help readers assess the dataset's representativeness and potential use cases. Could you explain the reasoning behind this decision and discuss any differences between the two periods that researchers should be aware of?

do you also log content features for the items? content features such as text/image/video/etc. content description (e.g., metadata, or embedding representations etc.) can be very useful features for recommendation.

**Details Of Ethics Concerns:**

this paper collected user data from an online recommendation system. The paper claim to have user consent and annoymized user-identity the technical preprocessing of the data (e.g., hashing etc.), but probably need double check privacy concerns when it's published.

---

> ### Author Response · Authors · 2024-11-25
>
> **Q1: The evaluation criterion for the quality of a benchmark dataset for industrial recommendation system should be fidelity to an actual online recommendation system. For example, if researchers come up with new algorithms with metrics improvement using this dataset, then when it’s deployed to a real online system, such improvement can be validated. So it would be great if the authors can demonstrate such fidelity to some extent, e.g., by running online A/B test to compare the online performance and offline metrics to see the correlation.**
>
> Thanks for your comments. The consistency between online and offline improvement is critical in industrial recommendation system. Here, we provide some cases to demonstrate the effectiveness of RecFlow. We have deployed FS-LTR [1] in the retrieval stage of our online recommendation system, resulting in a 0.18% improvement in watching time and a 0.69% increase in effective_view. Additionally, the implementation of FS-LTR [1] in the coarse ranking stage yielded a 0.22% improvement in watching time. In the ranking stage, we introduced DIN with an auxiliary ranking task, which enhanced watching time by 0.1%. These results from our online experiments not only confirm the effectiveness of the collected stage samples but also ensure consistency between our offline and online experiments.
>
> **Q2: The rules for determining how many samples for each stage seem quite ad-hoc w/o explanation of considerations. Could you explain the considerations that went into determining the number of samples collected at each stage? Are these numbers representative of typical production systems?**
>
> In the pre-ranking and coarse ranking stages, we have 8,000 and 3,000 samples, respectively, making it impractical to store all of them. Consequently, we randomly select 40 (10) filtered samples as negatives in both stages. Since the filtered videos in these stages are less relevant to user preferences, it is unnecessary to retain a large number of samples. In contrast, during the latter ranking and re-ranking stages, which are more aligned with user preferences, we aim to store as many samples as possible. Therefore, we retain 10 representative positive and negative samples in both stages during the first period. In the second period, we store all samples from these stages, providing a more accurate representation of a typical production system. Overall, the number of samples represents a balance between storage costs and representativeness.
>
> **Q3: What’s the rationale for partitioning the data collection into two periods? Understanding the rationale for partitioning the data collection into two periods would help readers assess the dataset's representativeness and potential use cases. Could you explain the reasoning behind this decision and discuss any differences between the two periods that researchers should be aware of?**
>
> We collect stage samples in this manner primarily to balance storage constraints with information integrity. The second period offers more comprehensive stage information compared to the first, providing researchers with greater flexibility to process the dataset according to their specific needs. In actual online data streams, we only collect samples using the approach implemented in the second period.
>
> **Q4: Do you also log content features for the items? content features such as text/image/video/etc. content description (e.g., metadata, or embedding representations etc.) can be very useful features for recommendation.**
>
> Thank you for your advice. As discussed in Section 4 of the Limitations, we currently do not log content features for items. We plan to include these features in the future.
>
> **Q5: Some typos: line 239: datatse -> dataset line 256: quote in wrong direction line 296/399: 1e-1/1e-2 not well formatted line 308/361: randomly sampling -> randomly sampled**
>
> We are sorry for those typos and will fix them.
>
> [1] Full Stage Learning to Rank: A Unified Framework for Multi-Stage Systems.

---

### Official Review · Reviewer_GxXY · 2024-10-28

**Soundness:** 3
**Presentation:** 4
**Contribution:** 4
**Rating:** 8
**Confidence:** 4

**Summary:**

The paper presents RecFlow, an industrial-scale recommendation dataset that captures the full recommendation pipeline with multiple stages, featuring 38M user interactions and 1.9B samples collected from 42K users and 9M items over a span of 37 days. One of RecFlow’s innovations is its inclusion of unexposed items at each pipeline stage, allowing for important analysis of distribution shifts between training and serving environments. The dataset also supports multi-task recommendation and behavior modeling by capturing various user feedback signals.

Experiments show that modeling stage-specific interactions and addressing distribution shift with RecFlow data improves recommendation performance, with some methods proving effective in real-world systems.

**Strengths:**

1. The paper presents the first comprehensive large-scale dataset that captures the complete recommendation pipeline, filling a critical gap in the field where existing datasets only contain exposure data. It could enable further research into real-world problems that were previously difficult to study, eg: distribution shift, stage interaction effects.
2. Good motivation is provided by clearly articulating the limitations of existing datasets and the importance of studying full recommendation pipelines.
3. The dataset is well documented with clear descriptions of features, collection methodology and privacy protection measures. The privacy protection approach is also robust, using a combination of user consent, feature anonymization and careful data filtering.
4. Experimental validation is thorough with multiple runs, standard deviation reporting and comprehensive ablation studies across different stages.

**Weaknesses:**

1. The paper doesn't adequately address the computational challenges of working with such a large dataset. Details about storage requirements and recommended sampling strategies would be valuable for practitioners.
2. The multi-task learning potential of the dataset is mentioned but not thoroughly explored. Given the rich set of user feedback signals, this seems like a missed opportunity.
3. While the authors mention online A/B testing validation, the details are sparse. More information about the production deployment and real-world performance would strengthen the paper's practical impact claims.
4. Analysis of how the stage samples could help with cold-start recommendations problem could be a useful contribution.

**Questions:**

1. What measures did you take to ensure the dataset is representative?
2. How can the dataset help handling cold-start users/items better?
3. Could you please provide more details on the online A/B testing setup and results?

---

> ### Author Response · Authors · 2024-11-25
>
> **Q1: Computational challenges, storage requirements, and recommended sampling strategies for working with such a large dataset.**
>
> RecFlow requires a total storage cost of 56 GB. A typical experiment, such as training the DIN ranking model, takes approximately two hours using an NVIDIA V100 with 32 GB of memory. Therefore, both storage and computational costs are manageable for researchers in the academic community. To further reduce resource expenses, we recommend utilizing data from February 5, 2024, to February 18, 2024.
>
> **Q2: The multi-task learning potential of the dataset is mentioned but not thoroughly explored.**
>
> Thank you sincerely for your feedback. Although we have discussed RecFlow's capabilities in multi-task recommendation, we did not initially present the results. Here, we provide the results of multi-task learning in RecFlow. RecFlow captures six types of user feedback: effective_view, long_view, like, follow, forward, and comment. We employ the widely recognized multi-task recommendation model MMoE [1] to learn these six types of feedback simultaneously. The table below presents the experimental results, along with comparisons to six single-task experiments.
>
> | Methods 	    | effective_view | long_view| like   | follow  | forward | comment |
> |-----------    |--------------  |----------|--------|---------|---------|---------|
> | Single Task   | 0.7270 	     | 0.7506 	| 0.9341 | 0.8699  | 0.8206  | 0.8659  |
> | MMoE[3]       | 0.7254 	     | 0.7500 	| 0.9309 | 0.8729  | 0.8144  | 0.8621  |
>
> The experimental results indicate that all tasks experience performance degradation, except for the "follow" task, highlighting the significant conflicts among tasks.
>
> **Q3: While the authors mention online A/B testing validation, the details are sparse. More information about the production deployment and real-world performance would strengthen the paper's practical impact claims. Could you please provide more details on the online A/B testing setup and results?**
>
> We appreciate your valuable suggestions for enhancing the robustness of our paper. We have deployed FS-LTR [2] in the retrieval stage of our online recommendation system, resulting in a 0.18% improvement in watching time and a 0.69% increase in effective_view. Additionally, the implementation of FS-LTR [2] in the coarse ranking stage yielded a 0.22% improvement in watching time. In the ranking stage, we introduced DIN with an auxiliary ranking task, which enhanced watching time by 0.1%. These results from our online experiments not only confirm the effectiveness of the collected stage samples but also ensure consistency between our offline and online experiments.
>
> **Q4: Analysis of how the stage samples could help with cold-start recommendations problem could be a useful contribution. And how can the dataset help handling cold-start users/items better?**
>
> Your questions consistently provide us with deeper insights into RecFlow. As discussed in JRC and BBP [3,4], the auxiliary ranking loss offers a more meaningful training signal for cold users with limited interactions, thereby enhancing their prediction performance. In sections A.5 and A.7 of the Appendix, we demonstrate our ability to design effective ranking losses based on stage samples. Additionally, ESAM [5] introduces a transfer learning framework to address the cold-start item recommendation problem by aligning representations between hot items and non-displayed items from stage samples. We anticipate that more methods focusing on cold-start user and item recommendations will be developed with the support of RecFlow in the future.
>
> **Q5: What measures did you take to ensure the dataset is representative?**
>
> On January 12, 2024, we randomly sample 42,000 seed users and record each recommendation request from January 13 to February 18, 2024. These sampled users exhibit diverse characteristics and are representative of the broader user base in our industrial recommendation system. The data collection process spans a total of 37 days, allowing RecFlow to capture a comprehensive profile of our recommendation system based on the recommendation requests from these 42,000 users. Furthermore, the collection methodology employed in RecFlow is comparable to that used in established recommendation datasets, including Tenrec [6] and KuaiRec [7].
>
>
> [1] Modeling Task Relationships in Multi-task Learning with Multi-gate Mixture-of-Experts.
>
> [2] Full Stage Learning to Rank: A Unified Framework for Multi-Stage Systems.
>
> [3] Joint Optimization of Ranking and Calibration with Contextualized Hybrid Model.
>
> [4] Beyond Binary Preference: Leveraging Bayesian Approaches for Joint Optimization of Ranking and Calibration.
>
> [5] ESAM: Discriminative Domain Adaptation with Non-Displayed Items to Improve Long-Tail.
>
> [6] Tenrec: A Large-scale Multipurpose Benchmark Dataset for Recommender Systems.
>
> [7] KuaiRec: A Fully-observed Dataset and Insights for Evaluating Recommender Systems.

---

### Official Review · Reviewer_p5Ha · 2024-10-31

**Soundness:** 2
**Presentation:** 3
**Contribution:** 3
**Rating:** 5
**Confidence:** 3

**Summary:**

This paper mainly focuses on introducing a dataset, RecFlow. This dataset contains full-flow recommendation data, including retrieval, pre-rank, coarse ranking, ranking, re-ranking, and edge ranking. Containing two periods, the datasets provide an opportunity to study full-stage recommendations in industry. The full-stage recommendation is widespread in industry recommendations and is supposed to be investigated. Some experiments are conducted to give examples of how to utilize this dataset.

**Strengths:**

1. An essential and practical problem in real industry recommendation. The full-stage recommendation is widespread in the industry; this dataset really provides a new perspective on this problem.

2. The collection strategy is provided, and privacy protection is carefully considered.

3. Experiments are provided to show how to use this dataset.

**Weaknesses:**

1. Despite providing collection and analysis, the collection procedure should be provided in more detail to show that it is reasonable and correct. Moreover, the analysis is too simple, and more intuition about this dataset can be given.

2. The experiments provided to show how to use this dataset are interesting. However, in line 079, the author argues that Recflow can provide merits of ten tasks. It should be supposed that the experiments on these tasks should be provided.

3. There are some typos. For example, Line 314 Recall@100,500,100 should be 1000. The whole paper should be proofread.

**Questions:**

1. Actually, the samples in every stage are based on the filtered strategy in the previous stage. So, will this strategy bring bias? And if we use a different strategy, can the conclusion still hold? For example, in industry, from retrieval to pre-ranking usually consists of several strategies. How does this benchmark reflect this?

2. Are the results from Table 4 to Table 7 reproducible?

---

> ### Author Response · Authors · 2024-11-25
>
> **Q1: Despite providing collection and analysis, the collection procedure should be provided in more detail to show that it is reasonable and correct. Moreover, the analysis is too simple, and more intuition about this dataset can be given.**
>
> Thank you for your review! The collection procedure is as straightforward as described in Section 2.1. We would be happy to provide additional details if the reviewer could specify which parts are unclear. The intuition behind this dataset is explained in lines 73-83. Specifically, RecFlow is the first dataset to include unexposed items in each request, reflecting the characteristics of an online recommendation system. This dataset can be utilized for various tasks, as outlined in lines 84-104, and can significantly advance research in recommendation systems. We will include more analysis, particularly comparing exposed and unexposed items, in future versions.
>
> **Q2: The experiments provided to show how to use this dataset are interesting. However, in line 079, the author argues that Recflow can provide merits of ten tasks. It should be supposed that the experiments on these tasks should be provided.**
>
> Thank you for your advice. The paper already highlights several merits of RecFlow. Here, we present additional results to further substantiate the other advantages of RecFlow.
>
> (1) Multi-task Recommendation
>
> Since RecFlow incorporates six types of user feedback, we demonstrate its application in multi-task recommendation. We first conduct six experiments with single-task models as the baseline, followed by an evaluation of the classical multi-task recommendation method, MMoE [1]. The experimental results presented in the table below indicate that all tasks experience performance degradation, except for the "follow" task, highlighting the significant conflicts among tasks.
>
> | Methods 	    | effective_view | long_view| like   | follow  | forward | comment |
> |-----------    |--------------  |----------|--------|---------|---------|---------|
> | Single Task   | 0.7270 	     | 0.7506 	| 0.9341 | 0.8699  | 0.8206  | 0.8659  |
> | MMoE[3]       | 0.7254 	     | 0.7500 	| 0.9309 | 0.8729  | 0.8144  | 0.8621  |
>
> (2) Context-based Recommendation
>
> RecFlow incorporates context, user, and video features in addition to identity features (e.g., user ID and video ID), making it well-suited for context-based recommendations. We replicate the classical method, FiBiNET [2], which adjusts feature importance based on the input context to enhance prediction accuracy. We compare FiBiNET with the baseline DIN, and the results presented in the following table demonstrate that context-based modeling can significantly improve performance.
>
> | Methods 	    | AUC    | LogLoss| Recall@50| NDCG@50 | Recall@100 | NDCG@100 |
> |-----------    |--------------   |----------|-------- |---------   |---------|---------|
> | DIN           | 0.7270 | 0.5828 | 0.1693   | 0.0449  | 0.3191     | 0.0691  |
> | FiBiNET[4]    | 0.7270 | 0.5821 | 0.1970   | 0.0526  | 0.3591     | 0.0787  |
>
> (3) Watching Time Prediction
>
> We implement the watching time prediction method using weighted logistic regression (WeightedLR) as proposed in YouTubeDNN [3]. We define positive and negative samples based on the effective_view label, where the weight of a positive sample corresponds to the user's watching time in seconds for the video, and the weight of a negative sample is set to 1. The loss function for WeightedLR is given by $loss=-w*ylog(\hat y)-(1-y)log(1-\hat y)$, where $w$ represents the watching time in seconds, $y$ is the effective_view label, and $\hat y$ is the model's prediction. During inference, we predict the watching time using $\hat{watching\\_time}=e^{logit}$, where $logit$ is the output of the last layer before the sigmoid activation function. We employ mean absolute error (MAE) as the performance metric. The results presented in the table below serve as a baseline for future research.
>
> | Methods 	    | MAE    |
> |-----------    |--------|
> | WeightedLR    | 22.75  |
>
> (4) Re-ranking
>
> In the re-ranking stage, we replicate the PRM [4] method. Consistent with the PRM framework, we select the top 30 ranked videos from the initial ranking stage as input for the re-ranking model. The input features include user_id, device_id, age, gender, province, photo_id, author_id, category_level_one, category_level_two, upload_type, upload_time, and request_time. The effective_view label is used to supervise the model's training. The PRM architecture comprises two layers of self-attention followed by one fully connected layer for prediction. The results presented in the table below serve as a baseline for future research.
>
> | Methods 	    | Prcession@1    |  Prcession@3    |  Prcession@6    |
> |-----------    |--------        |--------         |--------         |
> | PRM           | 8.31%          | 7.12%           | 6.18%           |

---

> > ### Author Response · Authors · 2024-11-25
> >
> > **Q3: There are some typos. For example, Line 314 Recall@100,500,100 should be 1000. The whole paper should be proofread.**
> >
> > We apologize for any typos that may have confused you. We will thoroughly review the paper and correct all errors.
> >
> >
> > **Q4: Actually, the samples in every stage are based on the filtered strategy in the previous stage. So, will this strategy bring bias? And if we use a different strategy, can the conclusion still hold? For example, in industry, from retrieval to pre-ranking usually consists of several strategies. How does this benchmark reflect this?**
> >
> > We also believe that the filtering strategy used in previous stages introduces a certain bias in online recommendation systems. Different strategies will indeed bring different bias and affect the distribution of the collected dataset, meaning our dataset inherently contains information about these filtering strategies. Since we design general algorithms/models based on the collected dataset, this bias is already accounted for, and the conclusions drawn from this dataset remain valid. In fact, the algorithm FS-LTR [5] has already been deployed in an online recommendation system, where filtering strategies vary over time. Despite these variations, the deployed method continues to show gains according to long-term backtesting experiments.
> >
> > **Q5: Are the results from Table 4 to Table 7 reproducible?**
> >
> > All results presented in the paper are fully reproducible. We have made the code, data, and software requirements available in our GitHub repository. Researchers who follow our instructions will be able to achieve the same experimental results. We will also rerun all experiments and provide checkpoints and training logs in the repository.
> >
> >
> > [1] Modeling Task Relationships in Multi-task Learning with Multi-gate Mixture-of-Experts.
> >
> > [2] FiBiNET: Combining Feature Importance and Bilinear feature Interaction for Click-Through Rate Prediction.
> >
> > [3] Deep Neural Networks for YouTube Recommendations.
> >
> > [4] Personalized Re-ranking for Recommendation.
> >
> > [5] Full Stage Learning to Rank: A Unified Framework for Multi-Stage Systems.

---

> > > ### Comment · Reviewer_p5Ha · 2024-11-26
> > >
> > > Thank you for your reply! My concern is still on how authors get samples from every stage with a specified filtered strategy. If the specified filtered strategy changes, will the dataset change accordingly? And will a proposed effective algorithm based on previous datasets become invalid? The whole paper will benefit from providing more details on this point.

---

> ### Author Response · Authors · 2024-11-26
>
> We appreciate the reviewer’s valuable question regarding the data collection process with a specific filtered strategy and the continual performance of algorithms in varying environments. Here are our perspectives:
>
> 1. Regarding the Data Collection Process with a Specific Filtered Strategy:
>
>     a. We employ a random sampling strategy within each stage to collect negative samples, as detailed in Section 2.1 and Fig. 1. For instance, in the pre-ranking stage, we randomly select 40 out of 5000 candidates as negatives (pre-rank_neg). By randomizing the sampling process, these 40 negative samples can approximately represent the data distribution in the pre-ranking stage, and similarly for other stages.
>
>     b. Since the dataset aims to represent a real-world multi-stage recommender system, it involves a specified filtering strategy: each stage only processes valid candidates that passed the previous stage. Such funnel-like filtering is common in real-world applications due to limited computing budgets. Note that such filtering and selection bias is widespread in datasets that only contain exposure samples, i.e., the best items filtered and displayed by the recommender. In contrast, our dataset collects samples from each stage of a real-world RS and adopts a random sampling policy within each stage. Therefore, compared with other datasets and algorithms that only utilize exposure samples, we believe the proposed dataset and algorithm can better eliminate such filtering bias and improve generalization performance.
>
> 2. Regarding the Continual Performance of Designed Algorithms in Varying Environments:
>
>     If the specified filtering strategy changes, the dataset will adjust accordingly. Learning an algorithm on a previous static/offline dataset would render it invalid. In contrast, our online RS collects data in a real-time data stream. As the filtering strategy evolves over time, this real-time data stream adapts correspondingly. Our deployed algorithm, being an online algorithm based on this real-time data stream, can thus recognize changes in the filtering strategy and adapt accordingly, ensuring its effectiveness. Additionally, as mentioned in our response to Q4, our deployed algorithm has demonstrated gains according to long-term backtesting experiments, during which the filtering strategy has changed.
>
> Furthermore, the filtering strategy is kept consistent during the period when collecting the offline dataset, ensuring it does not influence the offline experiments conducted on this dataset. This consistency ensures the dataset’s value for scientific research. Our algorithm (like FS-LTR), which performs well on this offline dataset, also shows gains in our online RS over the real-time data stream.
>
> We appreciate your valuable point again and will incorporate these details in the next version of our paper.

---

> > ### Comment · Reviewer_p5Ha · 2024-11-27
> >
> > Thank you for the explanation. My concerns still exist because the filtering strategy is different for every company, and it's not obvious how it affects algorithm design in research. I will keep my score, but I am okay with this paper being accepted.

---

> > > ### Author Response · Authors · 2024-11-27
> > >
> > > Thank you for your point! We agree that the filtering strategy varies for each company, directly influencing the released dataset. However, all existing datasets in the exposure space are also affected by their respective filtering strategies and differ from company to company. As we can see, these differences do not impede research in the recommendation systems (RS) area, including retrieval, ranking, and re-ranking. Some algorithms that perform well on offline datasets also work well in online environments. Our dataset can be seen as an extension of existing datasets from the exposure space to the un-exposure space, which we believe will aid in designing new algorithms for research. Additionally, although specific filtering strategies differ across companies, the data collection pipeline is generally consistent. We believe that a generally effective algorithm designed using such a dataset will also perform well across different companies, as these algorithms can be deployed within each company’s unique data stream.

---

### Official Review · Reviewer_UTjX · 2024-11-01

**Soundness:** 3
**Presentation:** 2
**Contribution:** 4
**Rating:** 6
**Confidence:** 4

**Summary:**

This paper first proposes a full-flow recommendation dataset collected from the industrial video recommendation scenarios. The overall process includes retrieval, pre-ranking, coarse ranking, ranking, re-ranking, and edge ranking. The logs are collected from January 13 to February 18, 2024. The datasets can be accessed via a half-anonymized link that denotes the authors' institute.

**Strengths:**

1. The proposed full-flow dataset provides a strong groundwork for follow-up research. For example, models can learn how to alleviate selection bias due to the discrepancy between the training and inference stages.
2. The authors performed comprehensive experiments and presented the results of the experiments with means and variances.
3. The complete datasets are available for further research.

**Weaknesses:**

1. The paper's current presentation lacks clarity and coherence, making it difficult to follow. Additionally, there are numerous minor grammatical and structural errors throughout the text.
2. While the initial explosion stage involves large-scale data, the subsequent re-ranking and edge-ranking stages utilize significantly smaller datasets. This inconsistency undermines the paper's claim of working with large-scale industrial data.
3. The paper's novelty is not effectively demonstrated through comparative analysis with existing work. Particularly in the introduction, while the authors enumerate the merits of the RecFlow dataset, they fail to provide meaningful comparisons with related work. The innovation of this research can only be discerned through prior knowledge of the field rather than through the authors' presentation.

**Questions:**

1. Regarding the ten merits presented in the introduction, it remains unclear which characteristics are unique to the RecFlow dataset compared to existing benchmarks.
2. As in line 143, what is the rationale behind the number of videos selected for each stage?
3. Also, can you explain why you chose 200 negative samples for each positive?
4. Some typos in the paper; for example, in line 379, recall 100 happens twice. This error occurs lots of times.

**Details Of Ethics Concerns:**

The proposed benchmark contains lots of users features, such as age, gender, province, etc.

---

> ### Author Response · Authors · 2024-11-25
>
> **Q1: The paper's current presentation lacks clarity and coherence. There are numerous minor grammatical and structural errors throughout the text.**
>
> Thanks for your review. We apologize for confusing you. The paper's core is why we collect the RecFlow dataset and how to utilize RecFlow. After rebuttal, we will revise the paper to make it clearer and coherent. Grammatical and structural errors will also be corrected.
>
> **Q2: While the initial explosion stage involves large-scale data, the subsequent re-ranking and edge-ranking stages utilize significantly smaller datasets. This inconsistency undermines the paper's claim of working with large-scale industrial data.**
>
> Thank you for your review! While it’s true that stages like re-ranking and edge ranking operate within a relatively smaller space compared to retrieval or ranking, the complexity of real-world systems far exceeds that of offline datasets. We believe that a large-scale industrial dataset can help bridge the gap between online and offline environments, facilitating the transfer of validated offline methods to real-world deployment. Additionally, releasing a large-scale industrial dataset provides researchers with the flexibility to choose between using a small or large dataset, as smaller datasets can be sampled from the larger one.
>
> **Q3: The paper's novelty is not effectively demonstrated through comparative analysis with existing work. Particularly in the introduction, while the authors enumerate the merits of the RecFlow dataset, they fail to provide meaningful comparisons with related work. The innovation of this research can only be discerned through prior knowledge of the field rather than through the authors' presentation.**
>
> The most significant novelty of RecFlow is that it includes filtered samples from each stage of the multi-stage industrial recommendation system, which is not present in existing recommendation datasets. We apologize for the lack of a comparative analysis with existing work in the introduction. However, we provide detailed comparisons with existing recommendation datasets in Section 2.4 (Comparison) and Appendix A.2 (Dataset Comparison). Our analysis shows that while some existing recommendation datasets possess certain advantages, RecFlow is the only dataset that encompasses all of these merits to date. We will include additional comparisons and highlight the innovations of this research in the introduction.
>
> **Q4: Regarding the ten merits presented in the introduction, it remains unclear which characteristics are unique to the RecFlow dataset compared to existing benchmarks.**
>
> Please see Section 2.4 (Comparison) and Appendix Section A.2 (Dataset Comparison) for more details. In detail: RecFlow has the following unique merit:
>
> (1) By recording items from the serving space, RecFlow enables the study of how to alleviate the discrepancy between training and serving for specific stages during both the learning and evaluation processes.
>
> (2) RecFlow also records the stage information for different stage samples, facilitating research on joint modeling of multiple stages, such as stage consistency or optimal multi-stage RS.
>
> (3) The rich information recorded about RS and user feedback allows the construction of more accurate RS simulators or user models in feed scenarios
>
> (4) Rich stage data may help estimate selection bias more accurately and design better unbiased algorithms.
>
> **Q5: As in line 143, what is the rationale behind the number of videos selected for each stage?**
>
> Currently, there is no established theory regarding the threshold number for each stage in industrial recommendation systems. The number of videos selected at each stage is manually determined based on model complexity, the number of machines available for inference, the number of candidates to be scored, and the results of online A/B tests. Designing better sampling strategies for each stage remains an area for future research.
>
> **Q6: Also, can you explain why you chose 200 negative samples for each positive?**
>
> When training the industrial retrieval model, the number of negative samples per positive sample significantly impacts performance, as initially stated in [1]. Our experiments demonstrate that increasing the number of negative samples from 1 to 200 enhances performance. However, using more than 200 negative samples per positive yields diminishing returns in terms of marginal benefit.
>
> **Q7: Some typos in the paper; for example, in line 379, recall 100 happens twice. This error occurs lots of times.**
>
> Thanks again for your careful review. The second figure should be 1,000 instead of 100. Therefore, the correct statement is: "For example, when we add 1 pre-ranked negative sample as a hard negative, the relative improvements in Recall@100, 500, and 1,000 are 46.8%, 42.4%, and 28.3%, respectively."
>
> [1] Embedding-based Retrieval in Facebook Search.

---

> > ### Comment · Reviewer_UTjX · 2024-11-30
> >
> > Thank you for your response. I think the authors should revise the typos and highlight them in the rebuttal version of the PDF.

---

> > > ### Author Response · Authors · 2024-12-02
> > >
> > > Thanks for your advice. Based on your constructive advice, we will improve our paper to make it clean and clear in the future version because the author can not upload a revised PDF after November 26th.

---

### Author Response · Authors · 2024-12-02

We sincerely thank all the reviewers for their time, effort, and constructive suggestions. We are pleased to know that all the reviewers agree to accept our paper. Although reviewer p5Ha maintains his/her score, he/she also supports the paper’s acceptance, stating, “I will keep my score, but I am okay with this paper being accepted.” All reviewers acknowledge the strong contribution of our proposed RecFlow dataset, with reviewers UTjX and GxXY rating it as excellent, and reviewers p5Ha and Y4hS rating it as good. They recognize RecFlow as foundational for future research in industrial multi-stage recommendation systems (reviewers UTjX, p5Ha, GxXY, and Y4hS). Additionally, reviewers appreciate the extensive experiments included in RecFlow (reviewers UTjX, p5Ha, and GxXY), the soundness of the methods (reviewers UTjX, p5Ha, GxXY, and Y4hS), and the quality of the paper’s writing (reviewers p5Ha, GxXY, and Y4hS). We have addressed each reviewer’s concerns and added the results for the requested experiments. We will continue to improve our paper by correcting typos and incorporating suggestions and critical points from the reviewers in future versions. Once again, we thank all the reviewers and encourage them to have further discussion as the rebuttal is ending.

---

### Meta-Review · Area_Chair_hwfw · 2024-12-21

**Metareview:**

This paper proposes a novel benchmark that includes unexposed items, which is a significant contribution to advancing research on multi-stage recommendation systems.  The results from online A/B testing also highlight its practical usefulness.
The authors have addressed many of the concerns in a clear and reasonable way.
Some questions still remain about generalizability, originality, and the thoroughness of the experiments, but the paper overall provides important insights that are worth sharing with the research community.

**Additional Comments On Reviewer Discussion:**

The authors have given reasonable answers to many of the main points.

---

### Decision · Program_Chairs · 2025-01-22

Accept (Poster)